# Effects of Aging on Adsorption of Tetracycline Hydrochloride by Humin

**DOI:** 10.3390/ijerph20042901

**Published:** 2023-02-07

**Authors:** Hongbo Hou, Guoliang Xu, Fei He, Hua Pan

**Affiliations:** 1Department of Resource and Environment, Baoshan College, Baoshan 678000, China; 2School of Energy and Power Engineering, Xi’an Jiaotong University, Xi’an 710049, China; 3Key Laboratory of Pollution Exposure and Health Intervention of Zhejiang Province, College of Biological and Environment Engineering, Zhejiang Shuren University, Hangzhou 310015, China

**Keywords:** adsorption, tetracycline, humin, aging, ferric hydroxide

## Abstract

To explore the effects of “aging”, an environmental factor, on adsorption of tetracycline hydrochloride (TC) by humin (HM), this paper coats HM surface with ferric hydroxide precipitate to simulate the aging process. The research findings indicate that compared with fresh HM, aged HM (HM-Fe) displays an accelerated adsorption rate and higher adsorption capacity on TC. With an initial concentration of 20 mg·L^−1^, TC’s equilibrium adsorption capacity on HM and HM-Fe is 4.6 and 5.3 mg·g^−1^, respectively, whereas the corresponding initial adsorption rate is 0.036 and 0.132 mg·g^−1^·min^−1^. The pseudo-second-order kinetic model and Freundlich adsorption isotherm model could adequately simulate the adsorption process of TC by HM and HM-Fe, suggesting the occurrence of chemical adsorption and multimolecular layer adsorption between TC and HM and HM-Fe. Based on ΔAbs deduced from Job’s calculation, it can be assumed a complex reaction occurs between the iron element on the HM-Fe surface and TC, which acts as a sort of bridge in strengthening the adsorption of TC by HM-Fe. The aforesaid findings may provide subsequent further study on environmental behaviors of TC in the soil with both fundamental theories and a scientific basis.

## 1. Introduction

Due to their inhibiting effect on microbial growth and reproduction, antibiotics have been widely applied in pharmaceutical and husbandry industries [1]. The average annual consumption of antibiotics in China is about 21 × 10^4^ t, which makes China the largest antibiotic-producing and antibiotic-consuming country across the world [2]. However, among the antibiotics used, about 54% may enter the soil [3]. The environmental behaviors of antibiotics in soil mainly consist of soil particle adsorption, adsorption by plant roots, microbial transformation, runoff transfer, and infiltration [4]. Among them, adsorption plays a key role in affecting the fate of antibiotics in the soil environment [4]. In addition, such physicochemical properties of soil as organic content, mineral composition, pH, and water content appear to be directly linked to antibiotic adsorption. Among those factors, soil organic matter (SOM) is commonly considered as a major influencing factor [5,6,7,8]. In recent years, numerous studies have claimed that the physicochemical properties of SOMs vary with the passage of time. This phenomenon is called “aging”. See and Bronk (2005) [9] extracted organic matter from sediment in rivers for an analysis, finding one-year-old SOMs to be different from the fresh ones in the quantity of O-H and N-H functional groups they carried. Yu et al. (2018) [10] pointed out that SOM aging was responsible for the lower pH. Thus, it can be speculated that aging materially affects the soil adsorption of pollutants [11,12,13]. Using pyrene, a member from the family of polycyclic aromatic hydrocarbons (PAHs), as the research subject, An et al. (2016) [14] concluded that the normalized distribution coefficient of soil organic carbon representing adsorption capacity *K_oc_* declined from 1465.28 L·kg^−1^ (in fresh SOM) to 848.64 L·kg^−1^ (in aged SOM) in their study.

Based on the aforementioned background, in this paper, tetracycline hydrochloride (TC) is selected as the absorbate. It is reported that daily antibiotic consumption in Britain is 21.2 DDD (daily defined dose) for every 1000 residents. Among the antibiotics consumed, TCs account for 22.1% [15]. As statistical data reveal, TC consumption in China in 2013 was as high as 1.2 × 10^5^ t. The TC residue in the soil environment of northern China was found to be 0.021–0.105 mg/kg by Hu et al. TCs are among the most frequently used antibiotics [16]. In addition, humin (HM) is selected as the absorbent here, because it is an important component of organic matter that is rich in oxygen-containing functional groups (e.g., hydroxy, carboxyl, and carbonyl) and can adsorb and thus bind with TCs [17,18,19]. This paper adopts the inorganic mineral coating method recommended by the International Humic Substances Society (IHSS) to simulate the aging process. To be more specific, ferric hydroxide is precipitated on the humin surface to compare the adsorption isothermal and kinetic performances of HM on TC before and after the aging process. It aims to provide a theoretical and scientific basis for subsequent further study on environmental behaviors of TCs in soil.

## 2. Materials and Methods

### 2.1. Soil Samples and Chemical Reagents

The soil samples used in this paper were collected from the Qinling Mountains at a geographical location of 109°15′36″ E and 33°55′12″ N. Soil was extracted from a thickness of 0–20 cm from the surface, adopting acheckerboard layout. All samples were mixed, naturally dried, ground, and screened (<2 mm) before being preserved in a −4 °C refrigerator for subsequent use. The physicochemical properties of the soil samples were reported by Cheng et al. (2020) [20].

Tetracycline (TC, purity > 98%) and sodium hydroxide (purity > 99%) were purchased from Dr. Ehrenstorfer (Germany), CaCl_2_ (purity > 96%) from Tianjin Tianli Chemical Reagents Co., Ltd. (China), FeCl_2_ (purity > 99%) from Tianjin Damao Chemical Reagents Factory (China), HCl solution (*w/v*: 38%) and HF solution (*w/v*: 40%) of analytical grade from J&K Technology Co., Ltd. (Shanghai, China). Purified water was prepared using a SPI-11-10T apparatus (ULUPURE, Sichuan, China).

### 2.2. Humin Extraction

The extraction of HM followed a previously described method [21]. A total of 100 g soil was added into a conical flask at 1:1 water-to-soil ratio. Hydrochloric (HCl) solution of three concentrations (0.1, 1.0, and 6.0 mol·L^−1^) was prepared. After the soil solution’s pH was regulated to 1.0 with 1 mol·L^−1^ of HCl solution, 0.1 mol·L^−1^ of HCl solution was added to fix mud to 400 mL and achieve 4:1 solid-to-liquid ratio. The mixed solution was vibrated on a 160 rpm shaker at 25 °C for 2 h and then centrifugated at 3000 rpm for 15 min before supernatant removal and collection and precipitation. A total of 1 mol·L^−1^ of sodium hydroxide (NaOH) was employed to centralize the precipitate pH to 7, and 0.1 mol·L^−1^ of NaOH was added under nitrogen protection to raise the solution volume to 400 mL. The collected precipitates were washed several times (at least five times) with 0.1 mol·L^−1^ NaOH. The solution was further vibrated on 160 rpm shaker for 24 h at 25 °C and centrifugated at 3000 rpm for 15 min. Afterwards, precipitate was collected and then dried at 60 °C and screened with a 2 mm sieve.

A 100 g sieved sample was placed into a 500 mL beaker. Three kinds of HCl solution with different concentrations of 1.0 mol·L^−1^, 6.0 mol·L^−1^ and 20 mol·L^−1^ were prepared. Into the beaker, 400 mL HCl (6 mol·L^−1^) was poured. The resulting solution was heated in a water bath at 60 °C for 20 h, and precipitate was collected after 15 min centrifugation at 3000 rpm was applied. The precipitate was washed with 16 mol·L^−1^ HCl and then centrifugated at 3000 rpm for 15 min to remove the supernatant. The procedure was repeated six times. Then, 165 mL of 6 mol·L^−1^ HCl and 335 mL of 22 mol·L^−1^ HF solutions were added, respectively. The mixed solutions were heated in a 60 °C water batch for 20 h and centrifugated at 3000 rpm for 15 min before being precipitated. The resulting precipitates were HM samples.

### 2.3. Humin Aging Processing

The humin aging processing followed a previously described method [22]. A 100 mL centrifuge tube containing a 0.5 g HM sample was added with a 0.1 mol·L^−1^ FeCl_2_ solution and then vibrated under 120 rpm for 24 h at 25 °C. A 20 mL NaOH (0.1 mol·L^−1^) solution was further added for 48 h of vibration. After that, the centrifuge tube was placed under 5 min centrifugation at 8000 rpm to remove the supernatant and subsequently collect the precipitate. The precipitate was washed repeatedly with purified water until the supernatant turned colorless. The resulting precipitate was then frozen and dried at −60 °C to process the aged HM sample before being preserved in a −4 °C refrigerator for subsequent use.

### 2.4. Adsorption Experiment

TC stock solution (1000 mol·L^−1^) was prepared and then diluted with a 0.01 mol·L^−1^ CaCl_2_ solution to generate stock solutions of different volumes and TC test solutions of different concentrations. Fresh and aged HM samples were chosen as absorbent.
(1)Adsorption kinetics: 0.3 g fresh and aged HM samples were weighed and placed into two 50 mL centrifuge tubes. The tubes were added with 30 mL TC solution (20 mg·L^−1^) and then sealed with Parafilm and vibrated under 120 rpm at 25 °C. Then, 10 mL samples were collected after a certain time to undergo 5 min vibration under 8000 rpm at 25 °C. Supernatant was collected and filtered with 0.45 μm film before being analyzed with a visible spectrophotometer (SP-195, Spectrum, Shanghai, China) at a 355 nm wavelength in order to determine TC concentration [4]. There were 3 parallel samples, and the results were averaged. Each absorbent was mixed with 0.01 mol·L^−1^ calcium chloride solution to form a control group to be processed under the same conditions as described above.(2)Adsorption isotherm: 0.3 g fresh and aged HM samples were taken and then placed into two 50 mL centrifuge tubes to be added with 20 mL TC solutions of different concentrations (in a range of 5–50 mg·L^−1^). Tubes were sealed with Parafilm and vibrated under 120 rpm at 25 °C for 24 h adsorption. Then, 10 mL samples were collected to undergo 5 min centrifugation under 8000 rpm. In the meanwhile, the supernatant was collected and filtered with 0.45 μm film to determine TC’s light adsorption at a 355 nm wavelength [23]. There were 3 parallel samples, and the results were averaged. Each absorbent was mixed with 0.01 mol·L^−1^ calcium chloride solution to form a control group to be processed under same conditions as described above.(3)Adsorption models: pseudo-first-order (1) and pseudo-second-order (2) kinetical models as well as a Weber and Morris intraparticle diffusion model (3) were employed to calculate kinetic parameters. Their mathematical formulas are shown below [7]:

(1)$$In\left(q_{e,exp}-q_{t}\right)=In\left(q_{e,cal}\right)-K_{1}t$$(2)$$\frac{t}{q_{t}}=\frac{1}{K_{2}q_{e,cal}^{2}}+\frac{1}{q_{e,cal}}t$$(3)$$qt=K_{id}t^{1/2}+I$$
where $q_{e,exp}$ represents equilibrium adsorption capacity (mg·g^−1^), $q_{t}$ represents adsorption capacity (mg·g^−1^) at moment t, $q_{e,cal}$ represents theoretical adsorption capacity (mg·g^−1^), $K_{1}$ represents the pseudo-first-order adsorption rate constant (min^−1^), $K_{2}$ represents the pseudo-second-order adsorption rate constant (g·mg^−1^·min^−1^), $K_{id}$ represents the intraparticle diffusion constant (g·mg^−1^·min^−1/2^), and I represents boundary layer thickness.

The adsorption isotherm was fitted using Freundlich and Langmuir models. Mathematical expressions of those two models are provided below [7]:(4)$$q_{e}=\frac{q_{m}K_{L}C_{e}}{1+K_{L}C_{e}}$$
(5)$$q_{e}=K_{F}C_{e}^{1/n}$$
where $q_{e}$ indicates equilibrium adsorption capacity (mg·g^−1^), $q_{m}$ indicates max adsorption capacity (mg·g^−1^), $K_{L}$ indicates the Langmuir constant (L·mg^−1^), $C_{e}$ indicates TC residue concentration in solution (mg·L^−1^), $K_{F}$ indicates the Freundlich constant (mg·g^−(1−1/n)^·g^−1^), and n indicates the Freundlich constant.

In this paper, the binding site between Fe^3+^ and TC was determined using Job’s computation. TC and Fe^3+^ were mixed at different ratios to have the resulting complex scanned with a spectrometer. In the meanwhile, TC and Fe^3+^ solutions of certain ratios received spectrum scanning separately. Here, light absorbance change (ΔAbs) refers to the sum of the Fe^3+^ solution’s light absorbance and TC solution’s light absorbance subtracted by their complex’s light absorbance. ΔAbs can be expressed with the following Formula (6) [22]:(6)$$\nabla \mathrm{Absorbance}=Abs\;\left(Fe^{3+}\right)+Abs\left(TC\right)-Abs\left(TC+Fe^{3+}\right)$$

## 3. Result and Discussion

### 3.1. Time for Achieving Adsorption Equilibrium

In this paper, fresh HM is called HM, whereas HM coated with 0.1 mol·L^−1^ ferric hydroxide is called HM+Fe. Figure 1a demonstrates the effect of contact time on TC adsorption by HM and HM+Fe. According to the figure, TC gets quickly adsorbed during the first 0–540 min, as the TC adsorption capacity ($q_{t}$) of HM and HM + Fe rises from 0 to 2.7 and 4.2 mg·g^−1^, respectively; $q_{t}$ rises at a much slower rate during the following 540–1440 min and remains almost unchanged during the subsequent 1440–3000 min. Based on $q_{t}$ change with t, 1440 min was selected as the time for achieving adsorption equilibrium, and adsorption capacity at this moment was determined as equilibrium adsorption capacity ($q_{e}$). The values of $q_{e,HM}$ and $q_{e,HM+Fe}$ are found to be 4.6 mg·g^−1^ and 5.3 mg·g^−1^, respectively, demonstrating that a higher amount of TC is adsorbed by HM+Fe. As shown by the illustration in Figure 1a, 48% TC has been adsorbed onto the surface of HM+Fe by 10 min, and the removal rate of TC under the equilibrium time is as high as 84%. Compared with HM+Fe, the HM surface has absorbed only 16% TC at 10 min, and the TC removal rate is as low as 52% even when the period is prolonged to the equilibrium time. In Figure 1b, the TC equilibrium removal rate by HM and HM+Fe at different concentrations is determined. It is found to be almost immune to the effect of the initial concentrations of TC ($C_{0})$. However, HM+Fe has an equilibrium removal rate twice that of TC under almost all concentrations. When the values of *C*_0_ were 5, 10, 20, 30, and 40 mg/L, the corresponding removal rates of TC over HM+Fe were calculated to be 79%, 81%, 84%, 82%, and 83%, whereas for HM they were 38%, 50%, 52%, 44%, and 47%, respectively. These results demonstrate that HM+Fe possesses a higher adsorption capacity towards TC compared to HM.

### 3.2. Adsorption Kinetic Model

Pseudo-first- and pseudo-second-order kinetic models were used to fit the curves shown in Figure 1a. Fitting results are provided in Figure 2 and Table 1. Compared with the pseudo-first-order kinetic model, the pseudo-second-order kinetic model has better fitness when applied to HM (R^2^ = 0.997) and HM-Fe (R^2^ = 0.996), and its $q_{e,cal}$ (being 4.650 and 5.263 mg·g^−1^ for HM and HM-Fe, respectively) is closer to the value of $q_{e,exp}$ (being 4.6 and 5.23 mg·g^−1^ for HM and HM-Fe, respectively). The higher applicability of the pseudo-second-order kinetic model indicates that TC absorbance by HM and HM-Fe is of a chemical type, which means the existence of electron sharing or exchange between them [24]. The $K_{2}q_{e,cal}^{2}$ in Table 1 indicates the initial adsorption rate [25], which is 0.132 and 0.036 mg·g^−1^·min^−1^ for HM+Fe and HM, respectively. The former is almost four-fold of the latter, suggesting quicker adsorption kinetic behavior.

The curves in Figure 1a were fitted with the Weber and Morris intraparticle diffusion model. The fitting results are as shown in Figure 3. In this figure, the first line segment represents the diffusion of TC from the solution body to the HM/HM-Fe outer surface, while the second one represents the diffusion of TC from the outer surface of HM/HM-Fe to the inner surface [26]. Because the first segment does not pass the origin point, it can be speculated that TC’s rate-limiting steps on HM/HM-Fe include both the boundary layer and intraparticle diffusion. Table 2 shows the parameter of Weber–Morris model.

### 3.3. Adsorption Isotherm Model

Figure 4 illustrates relatively high fitness of the Freundlich isothermal model when it is applied to TC adsorption by HM and HM+Fe. This suggests TC adsorption on those two substances is a kind of multimolecular layer adsorption. The parameter K_F_ means actual adsorption capacity, which is proven to be 0.440 and 1.187 mg·g^−(1−1/n)^·g^−1^ for HM and HM+Fe, respectively. It was observed from Figure 4 that under different initial concentrations, all the values of $q_{e,exp}$ were close to those of $q_{e,cal}$. The aforesaid results collectively indicate that HM+Fe could better adsorb TC. In summary, combined with adsorption kinetics and adsorption isotherm, it can be concluded that HM+Fe exhibits higher adsorption performance towards TC than HM.

### 3.4. Influence Mechanism of Aging on Adsorption

Aging normally has a negative effect on pollutant adsorption behavior. For example, as reported by Yu et al. (2018) [10], fresh humic acid (HA) could remove 88% of Cd. They placed HA in a greenhouse at 25/20 °C (day/night) to imitate the aging process and found that the Cd removal rate reached 46% after 130 days. An Xianjin et al. [14] coated HM with inorganic precipitated calcium carbonate (CaCO_3_) to simulate the aging process. It turned out that aged HM had a significantly lower performance in absorbing hydrophobic and organic pyrene when compared with the fresh HM. However, when they substituted calcium carbonate with ferric hydroxide precipitate (Fe(OH)_3_), contrary results were witnessed. This is consistent with our findings here. An Xianjin et al. attributed the change in organic pollutant–pyrene adsorption to the existence of the iron element [14]. The model pollutant used in this paper is tetracycline (TC) with a molecular structure as shown in Figure 5.

Zhao et al. [13] reported the possible binding reaction of ferric iron with TC. Figure 6a is a UV spectroscopy scanning of mixed ferric iron with TC at different ratios. On this basis, Job’s calculation was adopted to determine the ΔAbs value. The four rings in TC’s molecular structure, as illustrated in Figure 6b, are named from left to right as ring A, B, C, and D, respectively. All four rings could bind with ferric iron. Since ΔAbs has a wavelength range of 290–360 nm, we may determine that it is the oxygen atom (O) on ring D that appears reactive.

Based on the analysis above, it is believed that when ferric hydroxide precipitate (Fe(OH)_3_) is applied to coat the HM surface, the iron element may react with certain functional groups in HM so that the iron is partly resolved on the aged HM surface [14]. The iron element in a resolved state could further bind with TC to form a bridging effect so as to significantly enhance the TC adsorption rate and capacity. Consequently, the TC adsorption rate and capacity is improved accordingly.

## 4. Conclusions

In this paper, the humin aging process is simulated by coating the SOM-humin surface with an inorganic compound precipitate—ferric hydroxide. Compared with fresh HM, aged HM-Fe could adsorb TC at a quicker speed and with a bigger adsorption amount. The values of $q_{e,HM}$ and $q_{e,HM+Fe}$ are 4.6 mg·g^−1^ and 5.3 mg·g^−1^ at the equilibrium time, respectively. The applicability of the pseudo-second-order kinetic model here suggests that TC adsorption by HM and HM-Fe is of a chemical type. In this type of adsorption, adsorbate and adsorbent have electron sharing or exchange with each other. The values of $K_{2}q_{e,cal}^{2}$ are 0.132 and 0.036 mg·g^−1^·min^−1^ for HM+Fe and HM, respectively. The Freundlich adsorption isotherm model could well fit the adsorption process, indicating TC is adsorbed by HM and HM-Fe on multimolecular layers. The parameter K_F_ is proven to be 0.440 and 1.187 mg·g^−(1−1/n)^·g^−1^ for HM and HM+Fe, respectively. These results mean that HM+Fe presents higher adsorption performance towards TC than HM. The ΔAbs value computed based on Job makes it clear that TC could bind with the iron element on the HM-Fe surface to form a bridge. This also accounts for why aged HM could better adsorb TC. This study could shed some light on the further understanding of TC migration and transformation in the soil environment from both theoretical and scientific perspectives.

## Figures and Tables

**Figure 1 ijerph-20-02901-f001:**
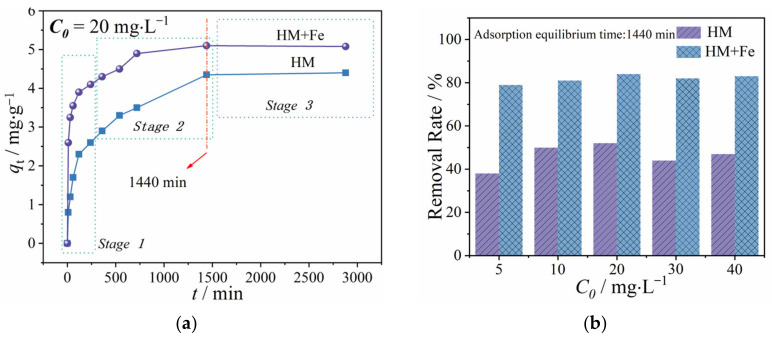
Effect of aging on the adsorption equilibrium and removal rate of TC by humic acid. (**a**) Effect of contact time on TC adsorption. (**b**) TC equilibrium removal rate under various initial concentrations.

**Figure 2 ijerph-20-02901-f002:**
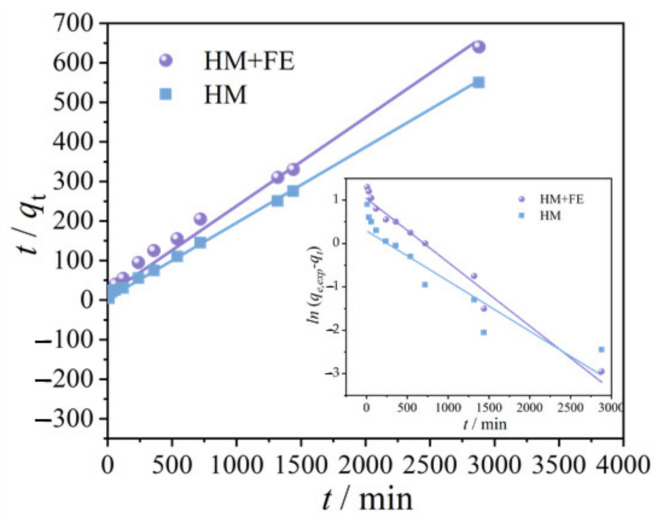
Pseudo-second-order kinetic models of TC on HM and HM+Fe (insert is the pseudo-second-order kinetic models).

**Figure 3 ijerph-20-02901-f003:**
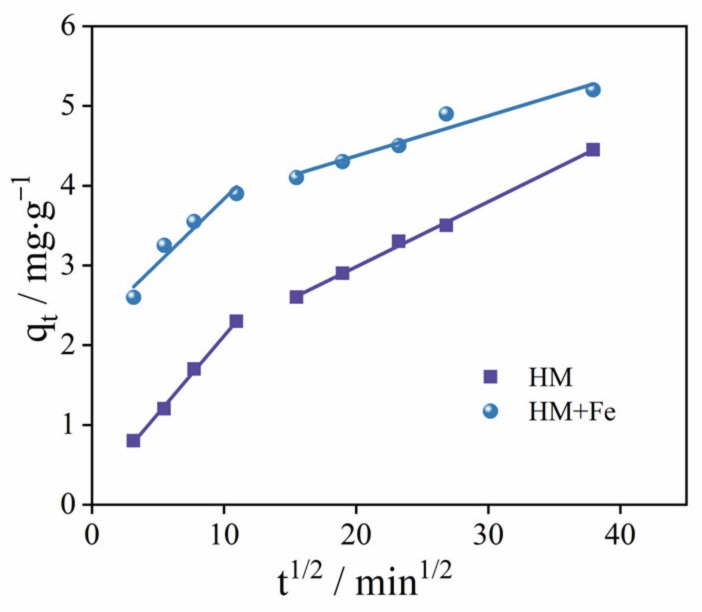
Weber–Morris intraparticle diffusion model of tetracycline on HM and HM+Fe.

**Figure 4 ijerph-20-02901-f004:**
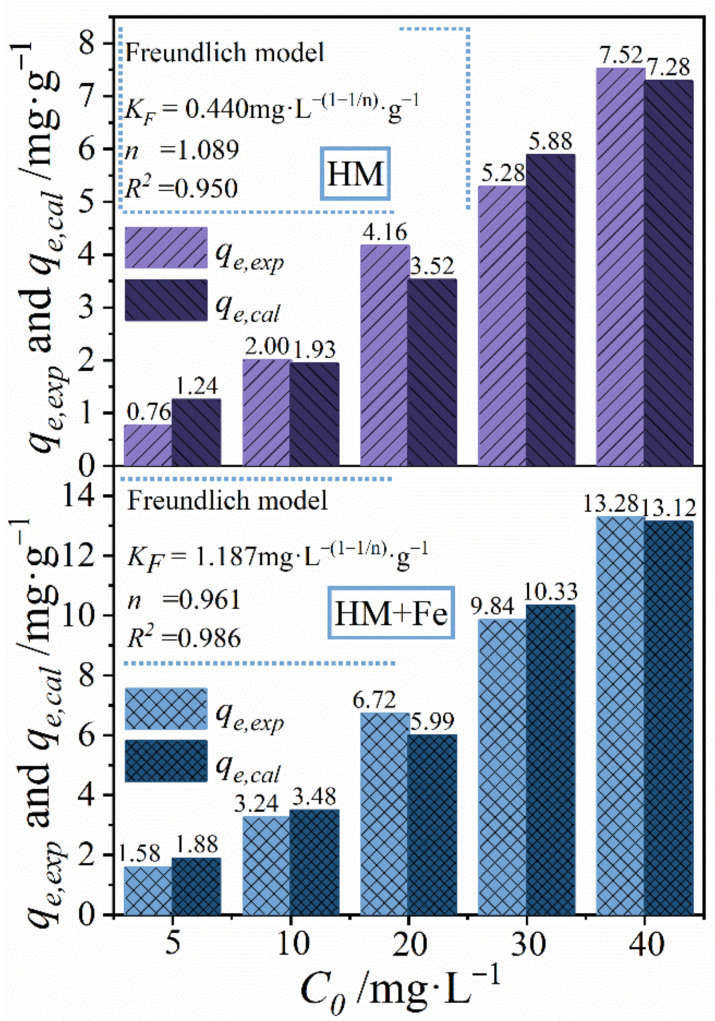
Parameters of adsorption isotherm model before and after HM aging.

**Figure 5 ijerph-20-02901-f005:**
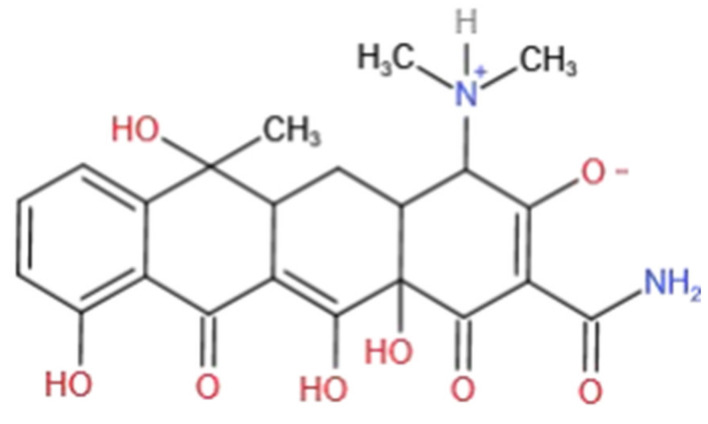
Molecular structural formula of TC.

**Figure 6 ijerph-20-02901-f006:**
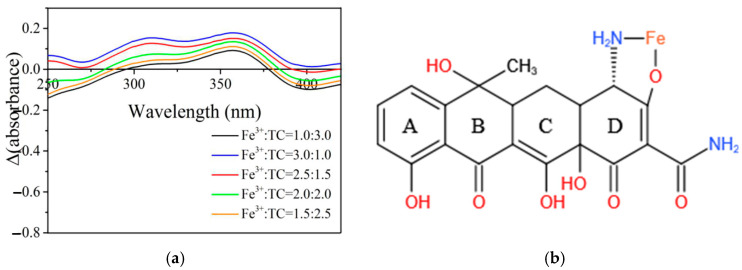
ΔAbs variation and possible molecular formula of its compound with Fe^3+^ and TC at different concentration ratios. (**a**) ΔAbs variation curve. (**b**) Molecular structure of TC-Fe compound.

**Table 1 ijerph-20-02901-t001:** Calculation of adsorption kinetics.

Model	Parameter	HM	HM+Fe
Pseudo-first-order kinetic model	*q_e,cal_*/mg·g^−1^	3.0	1.57
	*K*_1_/h^−1^	0.0015	0.0012
	*R* ^2^	0.974	0.861
	*K*_2_/g mg^−1^·min^−1^	0.002	0.005
Pseudo-second-order kinetic model	*q_e,cal_*/mg·g^−1^	5.263	4.650
	*K*_2_*q*^2^*_e,cal_*/mg·g^−1^·min^−1^	0.036	0.132
	*R* ^2^	0.997	0.996

**Table 2 ijerph-20-02901-t002:** Calculation of Weber–Morris model.

Model	Parameter	HM	HM+Fe
Weber–Morris intraparticle diffusion model	*K_id_* _1_/mg·g^−1^·min^−1/2^	0.1949	0.1611
	*I*_1_/mg·g^−1^	0.1682	2.2238
	*R* ^2^	0.9984	0.9426
	*K_id_* _2_/mg·g^−1^·min^−1/2^	0.0816	0.0503
	*I*_2_/mg·g^−1^	1.3507	3.3687
	*R* ^2^	0.9977	0.9460

## Data Availability

No new data were created or analyzed in this study. Data sharing is not applicable to this article.

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
