# Peer review of "Effects of Aging on Adsorption of Tetracycline Hydrochloride by Humin"

_ijerph, 2023, doi:10.3390/ijerph20042901_

Round 1
Reviewer 1 Report
Line 28: Due to the inhibiting effect on microbial growth and reproduction, antibiotics have 26 been widely applied in pharmaceutical and husbandry industries [1]. In 2013, total anti- 27 biotic consumption in China attained 162,000t, while its total yield was as high as 248,000t, 28 It makes China the largest antibiotic producing and consuming country across the world 29 [2], This data is very old, please use the recent data or add the recent contribution https://doi.org/10.1016/j.envpol.2021.117957.
Please highlight the role of Humin, and need of study, and cite recent literature like doi/10.1021/acsestengg.2c00118, DOI: 10.1016/j.envpol.2019.113678
Please change the heading of 2.2, abbreviation is not necessary.
Please improve or add more details of Figure 3 caption.
Please strengthen your methodology with some potential references.
Figure 5 caption is wrong, this is structural formula not molecular formula.
Figure 5 b, is it TIC scan ? This figure and caption is not clear.
There are number of grammar issue, please revise.
Author Response
- Line 28: Due to the inhibiting effect on microbial growth and reproduction, antibiotics have been widely applied in pharmaceutical and husbandry industries [1]. In 2013, total anti- biotic consumption in China attained 162,000 t, while its total yield was as high as 248,000 t, It makes China the largest antibiotic producing and consuming country across the world [2], This data is very old, please use the recent data or add the recent contribution https://doi.org/10.1016/j.envpol.2021.117957.
Response: Thank you very much for your professional advice, we have supplemented the research progress in the introduction section and added the literature according to the Reviewer’s comments, and all these changes are marked by red in the paper. And the revised text was listed as follows: “The average annual consumption of antibiotics in China is about 21×104 t, which makes China the largest antibiotic producing and consuming country across the world [2].
[2] Afzal Ahmed Dar, Bao Pan, Jian Qin, et al. Sustainable ferrate oxidation: Reaction chemistry, mechanisms and removal of pollutants in wastewater [J]. Environmental Pollution, 2021, 290, 117957.
- Please highlight the role of Humin, and need of study, and cite recent literature like doi/10.1021/acsestengg.2c00118, DOI: 10.1016/j.envpol.2019.113678
Response: Thank the reviewer for this insightful comment. We made modifications according to your suggestions. The correct was shown in line 53-55 in the revised manuscript. And the revised text was listed as follows: “In addition, HM is selected as the absorbent here, because it is an important component of organic matters that is rich in oxygen-containing functional groups (e.g., hydroxy, carboxyl, and carbonyl) and can adsorb and thus bind with TCs [17-19].
[17] QI Hui Mian, LV Liang, QIAO Xian Lian. Progress in Sorption of Antibiotics to Soils[J]. Soils, 2009, 41(5): 703-708.
[18] Afzal Ahmed Dar, Muhammad Usman, Wei Zhang, et al. Synergistic Degradation of 2,4,4′-Trihydroxybenzophenone Using Carbon Quantum Dots, Ferrate, and Visible Light Irradiation: Insights into Electron Generation/Consumption Mechanism[J]. 2022, 2: 1942-1952.
[19] A combined experimental and computational study on the oxidative degradation of bromophenols by Fe(VI) and the formation of self-coupling products [J]. Environmental Pollution, 2020, 258, 11678.
- Please change the heading of 2.2, abbreviation is not necessary.
Response: Thank you very much for the constructive suggestion. We have replaced “HM Extraction” by “Humin Extraction” in the revision.
- Please improve or add more details of Figure 3 caption.
Response: Thanks for the suggestion. We have replaced “Figure 3. Weber-Morris model of tetracycline on HM and HM+ Fe.” by “Figure 3. Weber-Morris intraparticle diffusion model of tetracycline on HM and HM+” in the revision.
- Please strengthen your methodology with some potential references.
Response: Thank you for your advice. We have added some important references in the revision.
- Figure 5 caption is wrong, this is structural formula not molecular formula.
Response: We are very sorry for our negligence. We have corrected this error in the revision. And the revised text was listed as follows: “Figure 5. Molecular structural formula of TC.”.
- Figure 5 b, is it TIC scan ? This figure and caption is not clear.
Response: We are very sorry. There wasn’t mentioned Figure 5 b in this study.
- There are number of grammar issue, please revise.
Response: We are very sorry for our negligence, and we have made careful corrections in the revision.
Reviewer 2 Report
Manuscript ijerph-2118715, "Effects of aging on adsorption of tetracycline hydrochloride by humin", by Hou et al, is interesting but it shows many flaws that impose a rejection.
The authors use Fe-enriched humin to compare their tetracycline absorption potential. With these results, they intend to explore the effect of humin “aging” and the variation of its absorption potential.
The HM extraction need to be revised. Some sentences are confusing and the names of solutions need to homogenize (better use HCl and HF). I am concerned about the extraction methodology. You do only one extraction with NaOH, usually more than one extraction is needed to extract all the humic acids. If these humic acids are not extracted, they will end up in the final product with humin. References are necessary in this section.
The last sentence of this section is confusing: “Then 6 molL-1 HCl and 22 molL-1 HF was added to get 165 mL and 335 mL solutions, respectively. Solutions were heated in 60 ℃ water batch for 20 h and centrifugated at 3000 rpm for 15 min before being precipitated. The resulting precipitates were HM samples.” Did you prepare two solutions at the end?
Another important aspect that needs to be developed is the “HM Aging Processing”. There is nothing about the chemical characterization of the material (Fe content in humin, chemical composition of humin,..). FeCl2 (Fe2+) was used, but the final results you mentioned Fe(OH)3 (Fe3+) in the humin, Why not Fe(OH)2 you need to explain this redox reaction. You need to prove the incorporation of the Fe in the organic structure in your final product. You need to add some references about the methodology in this section. Fe(OH)3 solubility is low in water, this compound can precipitate and be with the humin but not incorporate in the chemical structure of the humin.
In my opinion, it is necessary a deep characterization of the organic material to support all the results obtained in this study. The average of the results in triplicate must be with the standard deviations.
Author Response
- The HM extraction need to be revised. Some sentences are confusing and the names of solutions need to homogenize (better use HCl and HF). I am concerned about the extraction methodology. You do only one extraction with NaOH, usually more than one extraction is needed to extract all the humic acids. If these humic acids are not extracted, they will end up in the final product with humin. References are necessary in this section.
Response: Thank the reviewer for this insightful comment. We have homogenized the names of solutions according to your suggestions, and all these changes are highlighted by yellow in the paper. In addition, we have modified the extraction methodology according to the references. And the revised text was listed as follows: “1mol·L-1 sodium hydroxide (NaOH) was employed to centralize the precipitate pH to 7 and 0.1 mol·L-1 NaOH was added under nitrogen protection to raise solution volume to 400 mL. The collected precipitates were washed several times (at least five times) with 0.1 mol·L-1 NaOH.”
- The last sentence of this section is confusing: “Then 6 mol·L-1 HCl and 22 mol·L-1 HF was added to get 165 mL and 335 mL solutions, respectively. Solutions were heated in 60 ℃ water batch for 20 h and centrifugated at 3000 rpm for 15 min before being precipitated. The resulting precipitates were HM samples.” Did you prepare two solutions at the end?
Response: We are very sorry for our negligence. In fact, we prepared one solution at the end. We have made careful corrections in the revision. And the revised text was listed as follows: “Then added 165 mL of 6 mol·L-1 HCl and 335 mL of 22 mol·L-1 HF solutions, respectively. The mixed solutions were heated in 60 ℃ water batch for 20 h and centrifugated at 3000 rpm for 15 min before being precipitated.”.
- Another important aspect that needs to be developed is the “HM Aging Processing”. There is nothing about the chemical characterization of the material (Fe content in humin, chemical composition of humin.). FeCl2 (Fe2+) was used, but the final results you mentioned Fe(OH)3 (Fe3+) in the humin, Why not Fe(OH)2 you need to explain this redox reaction. You need to prove the incorporation of the Fe in the organic structure in your final product. You need to add some references about the methodology in this section. Fe(OH)3 solubility is low in water, this compound can precipitate and be with the humin but not incorporate in the chemical structure of the humin.
Response: Under anoxic conditions, Fe2+ in soil will form Fe(OH)2 under alkaline conditions. However, anoxic conditions were not set in this experiment, and Fe2+ are easy to be oxidized to form relatively stable Fe3+, then formed Fe(OH)3 under alkaline conditions[1]. In our previous study, it is found that Fe3+ will form complexes on the surface of humin. Therefore, the final results mentioned Fe3+ in the humin. Thanks for your advice, and In addition, we will conduct some characterizations of the organic material in the future.
[1] Wang H. Effects and mechanisms of Fe(Ⅱ/Ⅲ) on the degradation of tetracycline antibiotics in water [D]. Beijing: Beijing Jiaotong University, 2016:45-71.
- In my opinion, it is necessary a deep characterization of the organic material to support all the results obtained in this study.
Response: We agree with the recommendation and we think it’s significant to make a deep characterization of the organic material. In future studies, we will conduct SEM and other characterizations of the organic material.
- The average of the results in triplicate must be with the standard deviations.
Response: Thank you for the meaningful comment. The data points in our experiment are the average value of three times of repeated experiments, and the parallelism is good. Could you please allow us to make Figure by using the average value of data.
Reviewer 3 Report
The main idea of this study is clear but there are some questions and discussion need to be clarified as the following.
1.The first occurrence of "HM" in line 52 and "TC" in line 47 in "introduction" shall indicate the full name and abbreviation.
2.Line 67 "The physical properties of the soil samples were detailed in [18]." Suggest to put it another way.
3.3 and 4 suggestions are written as result and discussion.
4.There are too few references, and it is recommended to supplement.
5.It is suggested to adjust the English expression in many parts of the full text.
Author Response
- The first occurrence of "HM" in line 52 and "TC" in line 47 in "introduction" shall indicate the full name and abbreviation.
Response: Thank the reviewer for this insightful comment. We made modifications according to your suggestions, and all these changes are highlighted by red in the paper.
- Line 67 "The physical properties of the soil samples were detailed in [18]." Suggest to put it another way.
Response: We are very sorry for our negligence of language, and we have reconstructed this sentence.
- 3 and 4 suggestions are written as result and discussion.
Response: Thanks very much for your suggestion. We have changed the chapter of 3 and 4.
- There are too few references, and it is recommended to supplement.
Response: Thank you for your advice. We have added some important references in the revision.
- It is suggested to adjust the English expression in many parts of the full text.
Response: Thanks for the suggestion. The language was polished by a professional English writer as suggested.
Round 2
Reviewer 3 Report
In line 47, please change "tetracycline hydrochloride" to "tetracycline hydrochloride (TC)", and in line 53, "HM" to "human (HM)".
Author Response
1.In line 47, please change "tetracycline hydrochloride" to "tetracycline hydrochloride (TC)", and in line 53, "HM" to "human (HM)".
Response: Thanks for your careful review, and we have made these change in the revised manuscript.